# Distribution of 14 High-Risk HPV Types and p16/Ki67 Dual-Stain Status in Post-Colposcopy Histology Results: Negative, Low- and High-Grade Cervical Squamous Intraepithelial Lesions

**DOI:** 10.3390/cancers16193401

**Published:** 2024-10-05

**Authors:** Karolina Mazurec, Martyna Trzeszcz, Maciej Mazurec, Christopher Kobierzycki, Robert Jach, Agnieszka Halon

**Affiliations:** 1Corfamed Woman’s Health Center, Kluczborska 37, 50-322 Wroclaw, Poland; m.mazurec@corfamed.pl; 2Division of Pathology and Clinical Cytology, University Hospital in Wroclaw, Borowska 213, 50-556 Wroclaw, Poland; 3Division of Histology and Embryology, Department of Human Morphology and Embryology, Wroclaw Medical University, Chalubinskiego 6a, 50-368 Wroclaw, Poland; christopher.kobierzycki@umw.edu.pl; 4Division of Gynecologic Endocrinology, Jagiellonian University Medical College, Kopernika 23, 31-501 Krakow, Poland; jach@cm-uj.krakow.pl; 5Division of Clinical Pathology, Department of Clinical and Experimental Pathology, Wroclaw Medical University, Borowska 213, 50-556 Wroclaw, Poland; agnieszka.halon@umw.edu.pl

**Keywords:** cervical cancer screening, human papillomavirus, high-risk HPV, HPV genotyping, cervical intraepithelial neoplasia, p16/Ki67 dual staining, risk stratification

## Abstract

**Simple Summary:**

We studied how different types of high-risk human papillomavirus (HR-HPV) are distributed in early and advanced precancerous cervical lesions to better understand cervical cancer risks. A total of 559 cases were analyzed with results from a diagnostic test for HR-HPV and a special immunostaining method (DS) that helps identify cervical precancer potential that could enhance the effectiveness of the secondary cervical cancer prevention The investigation included a corresponding histology result from a colposcopic biopsy. The study found significant differences in the presence of HPV 16 and positive staining results between advanced lesions and those that were either negative or less severe. However, no significant difference was found between negative and less severe lesions. This study highlights the importance of understanding HPV type distribution and the use of dual staining in detecting and assessing the risk of cervical precancers.

**Abstract:**

Background: Determining the distribution of high-risk human papillomavirus (HR-HPV) types in histologic low-(LSIL) and high-grade (HSIL/CIN2+) squamous intraepithelial lesions through a diagnostic process in a cervical cancer prevention provides one of the key etiological factors behind further progression and persistence. Incorporating novel high-grade cervical lesion biomarkers such as p16/Ki67 dual staining (DS) alongside HPV typing has become important in detecting cervical precancers. Methods: Among 28,525 screening tests and 602 histology results, 559 cases with HR-HPV and histology results obtained from colposcopic biopsy were retrospectively analyzed, together with DS status. The χ^2^ test with Bonferroni correction evaluated the differences in HR-HPV type prevalence and DS positivity across three histologic study groups. Results: A statistically significant difference in the prevalence of HPV 16 was observed between negative and HSIL/CIN2+ (*p* = 0.00027) groups, as well as between the LSIL/CIN1 and HSIL/CIN2+ groups (*p* = 0.00041). However, no significant difference was found between the negative and LSIL/CIN1 groups. Similarly, the DS positivity difference was significant between the negative and HSIL/CIN2+ (*p* < 0.0001) and between the LSIL/CIN1 and HSIL/CIN2+ groups (*p* < 0.0001), but there was no significant difference between the negative and LSIL/CIN1 groups. Conclusions: The study highlights the heterogeneous nature of HPV-related cervical pathologies, and the distinct risks associated with different cervical lesion grades, emphasizing the importance of HR-HPV type distribution and DS status.

## 1. Introduction

Cervical cancer continues to be a major health challenge globally, being one of the most prevalent cancers affecting women worldwide [1]. It is predominantly caused by persistent infection with high-risk human papillomavirus (HR-HPV) types [2,3]. Among the over 200 human papillomavirus (HPV) genotypes identified, The International Agency for Research on Cancer (IARC) classified 13 types as carcinogenic or probably carcinogenic due to their strong association with the development of cervical precancer and cancer, with types 16 and 18 being responsible for the majority of cases worldwide [4,5,6,7].

The classification of cervical lesions into histologic low-grade squamous intraepithelial lesions (LSIL/CIN1) and histologic high-grade intraepithelial lesions with a quantification of cervical intraepithelial neoplasia in grade 2 or worse (HSIL/CIN2+: HSIL/CIN2, HSIL/CIN3 or HSIL unspecified) is crucial for determining the appropriate clinical management of patients during post-colposcopy surveillance or treatment after biopsy. LSIL/CIN1 is associated with high rates of spontaneous regression, while HSIL/CIN2+ has a higher potential to progress to cancer [8]. Identifying the distribution of HR-HPV types within precancerous lesions provides insight into the etiological factors driving the progression and persistence of these lesions. In addition to HPV typing, the integration of novel HSIL biomarkers such as p16/Ki67 dual staining (DS) in the detecting and prognosing precancers has gained prominence [9,10,11,12,13]. The DS enhances diagnostic accuracy by identifying cells that are simultaneously undergoing HPV-induced transformation and active proliferation. The immunocytochemical signal from two viral oncoproteins may indicate cell cycle dysregulation, leading to a transforming HPV infection, which can result in the development of high-grade cervical precancer or cancer [14,15]. In March 2020, the Food and Drug Administration approved DS for clinical use [16]. The Polish Society of Gynecologists and Obstetricians, together with the Polish Society of Colposcopy and Cervical Pathology, recommended DS for Polish cervical cancer screening, as the first ever worldwide guidelines, and in March 2024 DS was introduced into clinical use in the USA in the recommendations by the Enduring Consensus Guidelines Committee [10,17].

Understanding the baseline HPV infectious parameters can aid in improving the diagnostic precision, prognostic assessment, and management strategies for patients with cervical precancerous lesions. We herein retrospectively investigated the distribution of HR-HPV types by limited (partial) to types 16 and 18, extended genotyping and the DS status for negative colposcopic biopsy results and for low- and high-grade cervical squamous intraepithelial lesions and estimated the potential associations.

## 2. Materials and Methods

### 2.1. Study Population

This study is a post hoc analysis on liquid-based screening (LBS) tests, including HR-HPV, liquid-based cytology (LBC) and DS, in non-pregnant patients with a standardized colposcopy with biopsy performed. The data were collected from August 2015 to December 2022 at Corfamed Woman’s Health Center, one of the largest private outpatient gynecological clinics in Poland. The analysis covers 28,525 screening tests results and 602 histology results. Two cervical cancer screening strategies were used, namely primary cytology with reflex HPV testing and primary co-testing, in accordance with the Polish recommendations in force at that time [18]. Reflex HR-HPV testing was recommended for cases with minor cytological abnormalities [atypical squamous cells of undetermined significance (ASC-US) or low-grade squamous intraepithelial lesion (LSIL)]. A p16/Ki67 test was performed for all positive HR-HPV results. The final study group included 559 cases with HR-HPV and histology results from colposcopy with biopsy, aged 21 to 69 years old (mean age–35.4 years old), the selection process is presented in Figure 1. The study participants predominantly belonged to the middle to upper socioeconomic levels, with an advanced educational background, as all had completed at least secondary education. The endpoints of the study included a retrospective analysis of cytologic–virologic–immunocytochemical tests results, along with histology from colposcopies performed at the Center. All data were sourced from the Center’s electronic registry. The study was approved by the ethics committee (ID: 118.6120.36.2023). From 2020, the colposcopic protocol used in the study was recommended by The Polish Society of Colposcopy and Cervical Pathophysiology (PSCCP) as the optimal protocol [19]. Since the beginning of 2021, both screening strategies incorporating DS have been recommended for Polish conditions by the Polish Society of Gynecologists and Obstetricians and PSCCP [17].

### 2.2. Liquid-Based Screening (LBS), Sample Collection and Storage

The samples were collected with the Cervex-Brush device (Rovers Medical Devices, Oss, The Netherlands) and transferred into the BD SurePath vial (Becton Dickinson, Franklin Lakes, NJ, USA), according to the dedicated instructions for a collection, and this was used for all LBS tests, which included HR-HPV, LBC and DS, as specified above. The residual samples were stored under the conditions specified by manufacturers for 1–3 months, facilitating additional testing without additional patient appointments. All screening tests were processed, ensuring standardized procedures. The LBS tests (i.e., HR-HPV, LBC and DS) were conducted in two different laboratories—Abbott and Alinity with LBC and DS in one, and Onclarity with LBC and DS in another.

### 2.3. HR-HPV Testing with Two Assays for Extended Genotyping

Three molecular assays, including the Abbott RealTime High Risk HPV molecular in vitro PCR test (Abbott Molecular, Des Plaines, IL, USA), BD Onclarity HPV Assay (Becton Dickinson, Franklin Lakes, NJ, USA), and Alinity m HR HPV Assay (Abbott Molecular, Des Plaines, IL, USA), were performed to detect HR-HPV, all carried out in accordance with the manufacturers’ protocols. The Abbott test specifically phenotypes 12 HR-HPV types (31, 33, 35, 39, 45, 51, 52, 56, 58, 59, 66 and 68) and genotypes HPV 16 and 18 (limited genotyping), while the Onclarity (16, 18, 45, 31, 52, and 51 individually, and pooled results for 33/58, 35/39/68 and 59/56/66) and Alinity (16, 18, and 45 individually, and pooled results for 31/33/52/58 and 35/39/51/56/59/66/68) assays offer extended genotyping capabilities.

### 2.4. Liquid-Based Cytology and p16/Ki67 Dual-Stain Immunostaining

The cervical cytology samples were collected at the Center, placed in SurePath vials, and later processed following the manufacturers’ protocols in the automatic PrepStain Slide system (Becton Dickinson, Franklin Lakes, NJ, USA) in the external laboratories. The results were evaluated by a gynecological cytopathologist, informed of HR-HPV status, and according to the Bethesda 2014 system. The quality and control procedures followed the benchmarks from US laboratories accredited by the College of American Pathologists, with reporting rates in the study within established norms [20].

The CINtec PLUS detection kit (Roche, MTM AG laboratories, Munich, Germany) was used and processed in an automated BenchMark XT system (Ventana Medical Systems, Inc., Oro Valley, AZ, USA) for dual immunocytochemical staining with p16 and Ki67 proteins, according to the manufacturer’s instructions. The staining was performed using residual material from the original SurePath vials (Becton Dickinson, Franklin Lakes, NY, USA) stored in the laboratory after HPV and/or LBC testing. Each run included a control specimen. The p16/Ki67 slides were evaluated by a specially trained gynecological pathologist, who also assessed the original LBC samples. The results were categorized as positive, negative, or unsatisfactory based on specific criteria, including the presence of at least one cell with simultaneous red nuclear staining for Ki67 and brown cytoplasmic staining for p16, for a positive result [21].

### 2.5. Colposcopic Biopsy and Histology

According to the Polish recommendations, supplemented by the 2012 and 2015 American Society for Colposcopy and Cervical Pathology (ASCCP) guidelines, patients with p16/Ki67-positive, HPV 16/18-positive, ASC-US or LSIL HPV HR12-positive results, or cytologic atypical squamous cells, cannot exclude HSIL (ASC-H) or high-grade squamous intraepithelial lesion (HSIL), regardless of HR-HPV status, were referred for colposcopy with biopsy [18,22,23,24]. The colposcopy was conducted by Center colposcopists certified by the PSCCP. They followed a protocol that required at least endocervical sampling and directed biopsies for abnormal colposcopic findings, based on the 2011 International Federation for Cervical Pathology and Colposcopy (IFCPC) nomenclature. The random biopsies were taken if no abnormalities were found and the new squamocolumnar junction was visible, which was the obligatory procedure for major screening abnormalities. The histologic diagnoses of cervical biopsies and endocervical sampling were reviewed by the Center’s gynecological pathologist, using The Lower Anogenital Squamous Terminology (LAST) 2012 and World Health Organization (WHO)/IARC 2014/2020 terminology [25,26,27]. The colposcopies and histologic reports from outside the Center were excluded due to different colposcopic protocols, morphologic criteria, histologic nomenclature and/or no p16 immunohistochemistry performed. The reevaluation occurred when discrepancies between cytology, histology, and colposcopic findings were found, particularly when cytology indicated ASC-H or HSIL, but biopsies were negative or showed LSIL/CIN1. Multidisciplinary team meetings, both in person and online, were held to resolve these discrepancies.

### 2.6. Statistical Analysis

The χ^2^ test with Bonferroni correction was used to assess the differences in type prevalence and DS positivity between three groups of histology results. The analyses were performed using a 1.6.0 full version of licensed PQStat Software (2015 PQStat Statistical Calculation Software). The statistical significance was set at *p* < 0.05.

## 3. Results

The final study group included 559 cases with HR-HPV test and histology results. A total of 266 (47.6%) women had a negative histology result, 174 (31.1%) had a LSIL/CIN1 result and 119 (21.3%) had a HSIL/CIN2+ result. A total of 93.0% (520/559) of the patients were HR-HPV-positive and 7.0% were HR-HPV-negative. In the group with negative histology, HPV 16 was detected in 30.5%, HPV 18 in 7.5%, the 12 HR-HPV types other than HPV 16 and/or 18 (HR12) in 61.3% of cases, and 10.9% of patients were HR-HPV-negative. The most prevalent HR-HPV types among women with a LSIL/CIN1 result were HR12 (74.1%), followed by HPV 16 (29.3%) and HPV 18 (9.8%), and 4.0% of cases were HR-HPV-negative. In women with HSIL/CIN2+ results, the most common HR-HPV types were HR12 (58.0%) and HPV 16 (50.4%). HPV 18 was detected only in 5.9% of cases, and 2.5% were HR-HPV-negative. There was a statistically significant difference in the prevalence of HPV 16 between both the negative and HSIL/CIN2+ groups (*p* = 0.00027), as well as the LSIL/CIN1 and HSIL/CIN2+ groups (*p* = 0.00041), but there was no significant difference in the prevalence of HPV 16 between the negative and LSIL/CIN1 groups (*p* = 0.8816). The details are presented in Figure 2. The calculated HR-HPV-type positivity was non-hierarchical. The hierarchical analysis of HR-HPV positivity was also performed (a hierarchical design prioritizes type 16, followed by HPV 18, when the patient is negative for HPV 16, and subsequently HR12, when negative for both HPV types 16 and 18), and is shown in Figure 3 [28].

A single HPV 16 infection occurred in 22.6% of cases, and a multiple (HPV 16 and at least one HPV non-16) infection was noted in 8.0% of women with negative histology results. In the group with LSIL/CIN1 results, 16.1% of cases had a single HPV 16 infection and 13.1% had multiple. Meanwhile, in the cases with HSIL/CIN2+, a single HPV 16 infection was detected in 37.8% and a multiple infection in 12.6% of women. The detailed data are shown in Table 1.

The largest age group was the 30–39 group (224 cases) and the smallest groups were <25 (38 cases) and ≥50 (39 cases). The 25–29 group included 134 cases and the 40–49 group included 122 cases. In the <25 group, 34.2% of patients had a negative histology result, 47.4% had LSIL/CIN1 and 18.4% had HSIL/CIN2+. LSIL/CIN1 was found in 38.8% and HSIL/CIN2+ in 20.9% of women in the 25–29 group. A total of 26.3% of patients in the 30–39 group had a LSIL/CIN1 result and 27.7% had HSIL/CIN2+. In the 40–49 group, 30.3% of patients had a LSIL/CIN1 result and 15.6% had a HSIL/CIN2+ result, while in the ≥50 group, LSIL/CIN1 was found in 17.9% of cases and HSIL/CIN2+ in 5.1% (Figure 4).

In patients with histology results, HPV 16 was more prevalent in younger patients (<25, 25–29 and 30–39 years old)—36.8%, 41.0% and 36.2%, respectively. HPV 16 was detected only in 15.4% of women in ≥50 years of age. HPV 18 was the most common in the group of 25–29 years (10.4%), and the least frequent in women ≥50 (2.6%). The HPV HR12 were the most prevalent in the youngest (76.3%) and in the oldest (71.8%) patients. However, HPV HR12 were found in 60.7% of women in the 40–49 group and 61.2% in the 30–39 group (Figure 5).

A total of 430 (76.9%) patients in the final study group had an available DS result, and 61.9% of them (266) were DS-positive. In women with a negative histology result, 46.9% were DS-positive, 60.7% had a LSIL/CIN1 result and 92.9% had HSIL/CIN2+. The difference in the DS positivity between the negative and HSIL/CIN2+ (*p* < 0.0001) groups and the LSIL/CIN1 and HSIL/CIN2+ groups (*p* < 0.0001) was statistically significant. However, there was no significant difference in the prevalence of HPV 16 between the negative and LSIL/CIN1 groups (*p* = 0.0172) after applying the Bonferroni correction (significance level set at 0.0167). Among DS-positive patients with a negative histology result, HPV 16 was detected in 30.0% of cases, HPV 18 in 5.6% and HR12 in 56.7% of cases. The respective prevalences in women with LSIL/CIN1 were 27.1%, 10.6%, 71.8%, and in patients with HSIL/CIN2+, 47.3%, 4.4% and 60.4%. The details are presented in Figure 6.

## 4. Discussion

This study aimed to investigate the HR-HPV types distribution and the DS biomarker status in patients with negative, LSIL/CIN1 and HSIL/CIN2+ histology obtained in colposcopic biopsy. Our findings contribute to the understanding of the etiological factors driving the progression and persistence of these cervical lesions. We also highlighted the encouraging potential of integrating novel biomarkers into a clinical practice for an improved diagnostic accuracy by investigating the baseline HPV infectious patterns of different HR-HPV types through the limited and extended genotyping of two assays in relation to DS status. In our study, 47.6% of patients had a negative histology result in colposcopy, 31.1% had LSIL/CIN1, and 21.3% had HSIL/CIN2+ in histology. A total of 93.0% of women were HR-HPV-positive; the HR-HPV-negative cases were also included in our study, since they were referred to colposcopy based on either a ASC-H, HSIL and AGC cytology result or a DS-positive result. HPV 16 was the most frequently detected among histological diagnoses in the HSIL/CIN2+ group, underlining its critical role in the pathogenesis of high-grade lesions and cervical cancer. However, HPV HR12 and type 18 were most commonly found in the LSIL/CIN1 group, indicating the necessity for limited or extended HR-HPV-typing in the evaluation of cervical intraepithelial lesions. HPV 16 was more prevalent in younger patients, while infections with HPV HR12 were more common among both the youngest and the oldest patients and less frequent in the middle age-groups. The highest number of HSIL/CIN2+ lesions was noted in the 30–39 age group.

What is also worth emphasizing is that our results demonstrated that the incorporation of DS biomarker into the diagnostic workflow provided an additional diagnostic precision in distinguishing between productive HR-HPV infections and transforming infections with a higher risk of progression. The DS positivity rate was markedly higher in HSIL/CIN2+ (92.9%) lesions compared to LSIL/CIN1 (60.7%), reflecting the increased cellular proliferation and HPV-induced transformation in high-grade lesions.

Our results align with and expand upon the existing literature, offering both confirmations and novel observations regarding HR-HPV distribution and the utility of DS in the assessment of cervical intraepithelial lesions. In the study by Siegler et al., HPV 16 (41.5%) and 18 (3.7%) were slightly less prevalent than in our group in HSIL/CIN2+ patients (50.4% and 5.9%) [29]. In comparison to a study by Zeng et al., HPV 16 and 18 were detected more frequently than in our study in LSIL/CIN1 patients (HPV 16–29.3% vs. 12.6%; HPV 18–9.8% vs. 5.7%) and in HSIL/CIN2+ (HPV 16–50.4% vs. 34.1%; HPV 18–5.9% vs. 4.8%) patients [30]. The estimates presented in the study by Zhang et al. were lower for HPV 16 in the LSIL/CIN1 histologic results and HPV 18 in both the LSIL/CIN1 and HSIL/CIN2+ groups, but similar for HPV 16 in the latter group (45.7% vs. 50.4% in our study) [31]. Similarly to the studies by de Sanjose et al. and Guan et al., we identified HPV 16 as the predominant type associated with high-grade lesions [32,33]. Clifford et al. also reported a diverse distribution of HR-HPV types in low-grade lesions, suggesting that other HR-HPV types play significant roles in the early stages of cervical neoplasia [34]. Our observation that HPV 16 was more prevalent in younger patients, while other HPV HR12 types were more common among both the youngest and the oldest patients, aligns partially with research by Bruni et al., who noted variations in HR-HPV-type prevalence across different age groups [35]. The higher prevalence of HPV 16 among younger patients could be attributed to the higher sexual activity in this demographic, while the distribution pattern of HR12 types may reflect differing transmission dynamics and immune responses in different age cohorts. The finding that DS positivity was higher in HSIL/CIN2+ compared to LSIL/CIN1 is corroborated by a study from Wentzensen et al. [36]. Overall, we have observed similar results to other studies, with slightly higher HPV 16 and 18 prevalences. As previously observed, HPV 16 was more common in HSIL/CIN2+ cases than in LSIL/CIN1 and negative histology results, and more women with HSIL/CIN2+ than with LSIL/CIN1 histology results from colposcopic biopsy were DS-positive.

This study highlights several strengths. Notably, it offers the largest non-interventional analysis of LBS results in Poland and Central Eastern Europe, providing comprehensive insights into private opportunistic cervical cancer screening results and covering a wide age range of participants. Additionally, the study stands out as one of the largest investigations into cytologic–virologic–immunocytochemical–histologic correlations in cervical cancer screening, with all LBC and DS evaluated by a qualified gynecologic cytopathologist. However, the limitations include its retrospective nature and the exclusion of colposcopic biopsy results from outside facilities due to differing protocols and histologic terminology. Conducted in a single private funds-based center and in a relatively homogeneous study group, the results may not be generalizable to other populations.

## 5. Conclusions

The HR-HPV types distribution and the p16/Ki67 DS status in cervical squamous intraepithelial lesions in samples obtained in colposcopic biopsy underscore the heterogeneity of HPV-related cervical pathologies and the varying risks associated with different lesion grades. HPV 16 remains the most significant type in HSIL/CIN2+, reaffirming its central role in cervical oncogenesis. The higher prevalence of DS positivity in HSIL/CIN2+ highlights the potential of this biomarker in improving diagnostic precision and aiding in the stratification of patients based on their risk of progression. These findings emphasize the need for a comprehensive approach in the management of cervical squamous intraepithelial lesions, integrating HR-HPV genotyping (limited to types 16 and 18, or extended) and novel biomarkers such as DS to enhance the accuracy of diagnosis and the effectiveness of treatment strategies. Further research should focus on longitudinal studies to validate these biomarkers’ prognostic value and to develop optimized protocols for the management of cervical intraepithelial lesions, ultimately aiming to reduce the burden of cervical cancer through early and precise intervention.

## Figures and Tables

**Figure 1 cancers-16-03401-f001:**
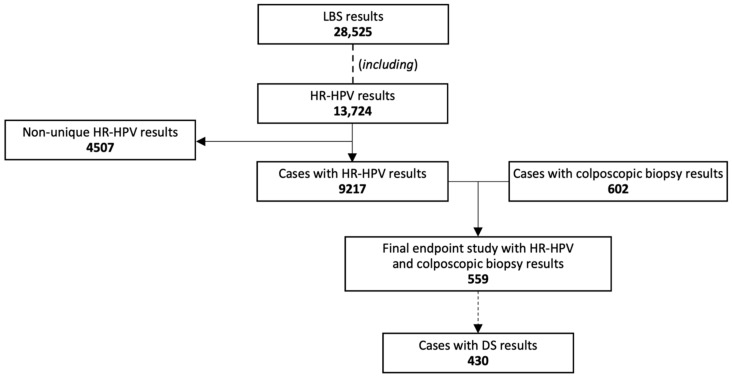
The study design process. Abbreviations: LBS, liquid-based screening; HR-HPV, 14 high-risk types of human papillomavirus test; DS, p16/67 dual-stain testing.

**Figure 2 cancers-16-03401-f002:**
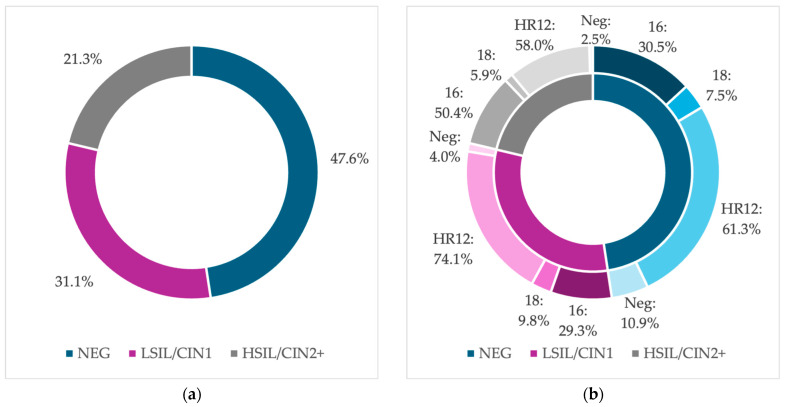
The histology results (**a**) with non-hierarchical HR-HPV types distribution (**b**). Abbreviations: NEG, negative histology result; LSIL/CIN1, histologic low-grade squamous intraepithelial lesion; HSIL/CIN2+, histologic high-grade intraepithelial lesion with a quantification of cervical intraepithelial neoplasia in grade 2 or worse; HR-HPV, 14 high-risk types of human papillomavirus; 16, positive results for human papillomavirus type 16; 18, positive results for human papillomavirus type 18; HR12, positive results for 12 high-risk types of human papillomavirus, other than types 16 and 18; Neg, human papillomavirus negative results.

**Figure 3 cancers-16-03401-f003:**
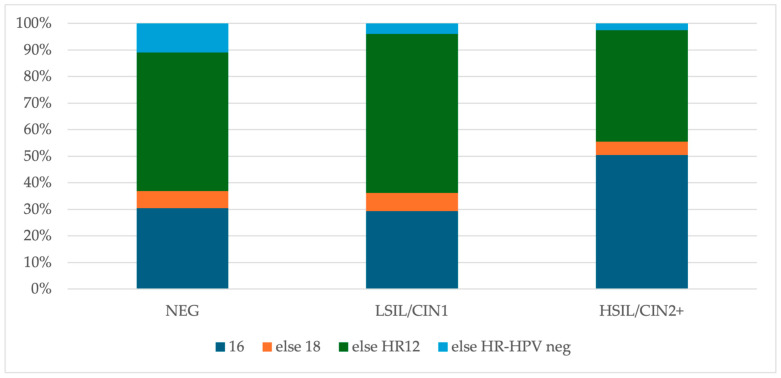
The hierarchical HR-HPV types distribution in histology results. Abbreviations: NEG, negative histology result; LSIL/CIN1, histologic low-grade squamous intraepithelial lesion; HSIL/CIN2+, histologic high-grade intraepithelial lesion with a quantification of cervical intraepithelial neoplasia in grade 2 or worse; HR-HPV, 14 high-risk types of human papillomavirus; 16, positive results for human papillomavirus type 16; 18, positive results for human papillomavirus type 18; HR12, positive results for 12 high-risk types of human papillomavirus, other than types 16 and 18; neg, negative results for human papillomavirus; the hierarchical design prioritizes type 16, else HPV 18 when patient is negative for HPV 16, else HR12 when negative for both HPV types 16 and 18.

**Figure 4 cancers-16-03401-f004:**
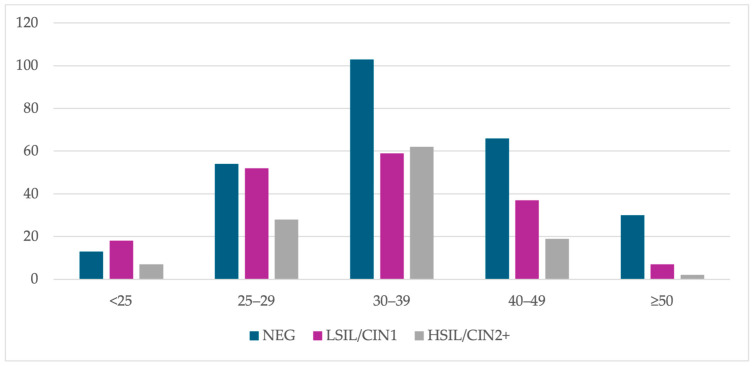
The age-specific number of cases with histology results from colposcopic biopsy. Abbreviations: NEG, negative histology result; LSIL/CIN1, histologic low-grade squamous intraepithelial lesion; HSIL/CIN2+, histologic high-grade intraepithelial lesion with a quantification of cervical intraepithelial neoplasia in grade 2 or worse.

**Figure 5 cancers-16-03401-f005:**
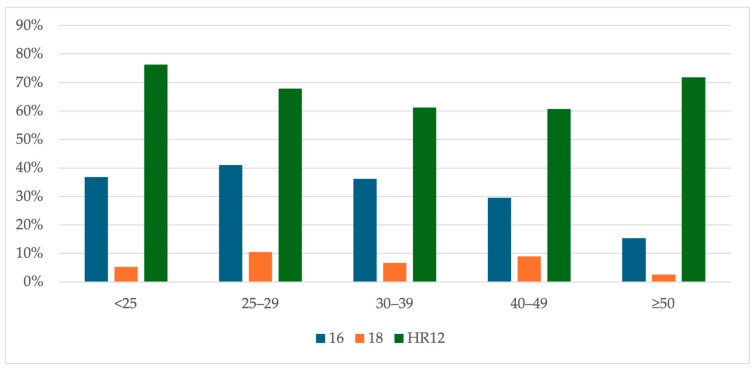
The age-specific HR-HPV types distribution (non-hierarchical) in patients with histology results from colposcopic biopsy. Abbreviations: HR-HPV, 14 high-risk types of human papillomavirus; 16, positive results for human papillomavirus type 16; 18, positive results for human papillomavirus type 18; HR12, 12 high-risk types of human papillomavirus other than types 16 and 18 positive results.

**Figure 6 cancers-16-03401-f006:**
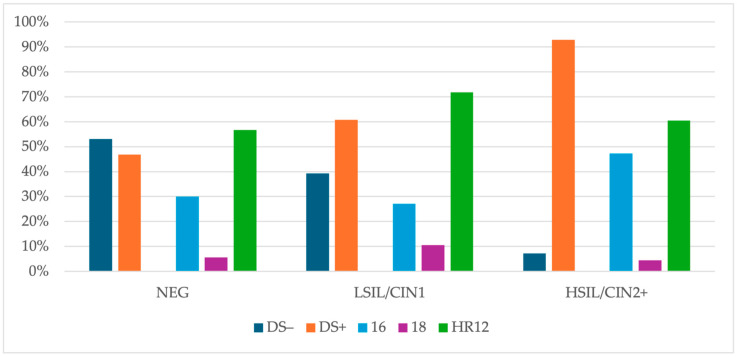
The DS results and HR-HPV types distribution (non-hierarchical) in DS-positive cases. Abbreviations: HR-HPV, 14 high-risk types of human papillomavirus; NEG, negative histology result; LSIL/CIN1, histologic low-grade squamous intraepithelial lesion; HSIL/CIN2+, histologic high-grade intraepithelial lesion with a quantification of cervical intraepithelial neoplasia in grade 2 or worse; DS, p16/67 dual-stain testing; +, positive; −, negative; 16, positive results for human papillomavirus type 16; 18, positive results for human papillomavirus type 18; HR12, positive results for 12 high-risk types of human papillomavirus, other than types 16 and 18.

**Table 1 cancers-16-03401-t001:** The characteristics of HR-HPV infections in post-colposcopy histology results.

Positive	NEG (n = 266), No (%)	LSIL/CIN1 (n = 174), No (%)	HSIL/CIN2+ (n = 119), No (%)
HR-HPV	237 (89.1)	167 (96.0)	116 (97.5)
HPV 16	60 (22.6)	28 (16.1)	45 (37.8)
HPV 18	12 (4.5)	7 (4.0)	1 (0.8)
HPV 16 and 18	2 (0.8)	3 (1.7)	1 (0.8)
HPV 16 and HR12	18 (6.8)	18 (10.3)	14 (11.8)
HPV 16, 18 and HR12	1 (0.4)	2 (1.1)	0 (0.0)
HPV 18 and HR12	5 (1.9)	5 (2.9)	5 (4.2)
HR12	139 (52.3)	104 (59.8)	50 (42.0)
Negative	29 (10.9)	7 (4.0)	3 (2.5)

Abbreviations: NEG, negative histology result; LSIL/CIN1, histologic low-grade squamous intraepithelial lesion; HSIL/CIN2+, histologic high-grade intraepithelial lesion with a quantification of cervical intraepithelial neoplasia in grade 2 or worse; HR-HPV, 14 high-risk types of human papillomavirus; HPV 16, human papillomavirus type 16; HPV 18, human papillomavirus type 18; HR12, 12 high-risk types of human papillomavirus other than types 16 and 18.

## Data Availability

The data presented in this study are available upon reasonable request to the corresponding authors.

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
