# Peer review of "Distribution of 14 High-Risk HPV Types and p16/Ki67 Dual-Stain Status in Post-Colposcopy Histology Results: Negative, Low- and High-Grade Cervical Squamous Intraepithelial Lesions"

_cancers, 2024, doi:10.3390/cancers16193401_

Round 1

Reviewer 1 Report

Comments and Suggestions for Authors

The authors aimed to retrospectively analyze the HR-HPV type distribution and the p16/Ki67 staining status in patients with negative, LSIL/CIN1 and HSIL/CIN2+ histology. A large body of data has been analyzed, on the basis of which conclusions could be drawn about how different testing methods complement each other and what contribution they can make to the overall reliability of screening.

However, the analysis presented in the manuscript does not provide an opportunity to fully draw such conclusions and is purely descriptive. Some identified patterns are well known (a statistically significant difference in the HPV 16 prevalence between different lesion grades, age-dependent prevalence of different HPV types, the higher prevalence of DS positivity in HSIL patients). The other conclusions are insufficiently supported.

I.e., the conclusion “our results demonstrated that the incorporation of DS biomarker into the diagnostic workflow provided an additional diagnostic precision in distinguishing between productive HR-HPV infections and transforming infections with a higher risk of progression” (lines 284-287) is not supported by the presented results, since no information is provided on the further progression (or non-progression) of the disease in patients depending on their DS status. Thus, the authors cannot judge that with DS they distinguished productive from transformative infections. The conclusion “However, HPV HR12 and type 18 were most commonly found in the LSIL/CIN1 group, indicating the necessity for comprehensive HR-HPV typing in the evaluation of cervical intraepithelial lesions” (lines 277, 278) is unfounded. In fact, the results presented illustrate the limited informativeness of partial HPV genotyping for cervical screening purposes.

Perhaps a more detailed representation of information about patients and samples in the Materials and Methods section (i.e., duration of sample storage, detailed age stratification (since the expression of p16ink4a increases with age), HPV viral loads (or, at least, HPV-specific signal intensities), cytological findings, etc) would allow the authors to identify additional patterns in the distribution of the results of different analyses by subgroups of patients. I.e., importantly, no significant difference in the prevalence of HPV 16  and DS-positivity rate was found between negative and LSIL/CIN1 groups. But was there concordance between DS-positivity and HPV16 positivity? Have the authors identified DS-positive but HPV-negative cases?

In my opinion, of particular interest is the heterogeneity of "histologically negative" patients with respect to their DS status and the high proportion of DS-positives in this cohort. However, this group of patients and this result is discussed quite superficially in the manuscript. There is a lack of detailed information about the reasons for referring patients from these subgroups for colposcopy and biopsy, about previous cytological conclusions. It is possible that DS-positive and DS-negative patients from this cohort differed among themselves and in other characteristics. Also to my view, it could be interesting for the reader to analyze the DS status of the samples for which discrepancies were obtained between different methods of testing, i.e. between cytology, histology, and colposcopic findings. Such an analysis could help point to possible causes and sources of discrepancies.

At Figure 2, section a) looks redundant, since the same data are presented in section b).

The text contains a number of easily removable typos and stylistic errors, i.e., “The most prevalent HR-HPV among women with LSIL/CIN1 result were HR12” (while there is no HPV variety like HR12).

My general impression - the article can be published after some revision, including more detailed representation of results as well as reformulating some of the conclusions.

Comments on the Quality of English Language

I have no comments on the language

Author Response

Response to Reviewer 1 Comments

(Round 1)

1. Summary

We would like to thank very much Reviewer 1 for taking the time to review this manuscript and the valuable help. We greatly appreciate the insights, and we have carefully considered each of the comments. Please find the detailed responses below and the corresponding revisions highlighted in yellow in the re-submitted file.

2. Point-by-point response to Comments and Suggestions

Comments 1: The authors aimed to retrospectively analyze the HR-HPV type distribution and the p16/Ki67 staining status in patients with negative, LSIL/CIN1 and HSIL/CIN2+ histology. A large body of data has been analyzed, on the basis of which conclusions could be drawn about how different testing methods complement each other and what contribution they can make to the overall reliability of screening.

Response 1: We would like to thank Reviewer 1 for their thoughtful comments. We appreciate the recognition of the scope of data analyzed and agree that our study provides valuable insights into how various testing methods complement each other, enhancing the reliability of screening. This was indeed a key objective of our research.

Comments 2: However, the analysis presented in the manuscript does not provide an opportunity to fully draw such conclusions and is purely descriptive. Some identified patterns are well known (a statistically significant difference in the HPV 16 prevalence between different lesion grades, age-dependent prevalence of different HPV types, the higher prevalence of DS positivity in HSIL patients).

Response 2: We thank Reviewer 1 for these insightful comments and approach. The primary objective of our study was to determine whether findings previously observed in European and global studies are applicable in a Polish context, where such data has not been extensively reported. Thus, the above-mentioned HPV 16 prevalence in different cervical precancer detections represents new data for the Polish region. Our study allows us to confirm the consistency of the prevalence reported in other published studies within the analyzed context. There is limited Polish data regarding HPV prevalence in specific cervical precancers using p16 in histological diagnosis, and no data where patients who underwent p16/Ki67 testing have been included. Additionally, there is limited global evidence on the latter issue. A description of the study group can be found in subsection 2.1 of the Materials and Methods, as well as in the study design process presented in Fig. 1.

Comments 3: The other conclusions are insufficiently supported. I.e., the conclusion “our results demonstrated that the incorporation of DS biomarker into the diagnostic workflow provided an additional diagnostic precision in distinguishing between productive HR-HPV infections and transforming infections with a higher risk of progression” (lines 284-287) is not supported by the presented results, since no information is provided on the further progression (or non-progression) of the disease in patients depending on their DS status. Thus, the authors cannot judge that with DS they distinguished productive from transformative infections. The conclusion “However, HPV HR12 and type 18 were most commonly found in the LSIL/CIN1 group, indicating the necessity for comprehensive HR-HPV typing in the evaluation of cervical intraepithelial lesions” (lines 277, 278) is unfounded. In fact, the results presented illustrate the limited informativeness of partial HPV genotyping for cervical screening purposes.

Response 3: We thank Reviewer 1 for this valuable observation. However, we would like to clarify that p16/Ki67 dual staining (DS) is a biomarker indicating the risk of high-grade lesions, which is also associated with an elevated risk of transforming HPV infections [Reference 14: Wentzensen N, von Knebel Doeberitz M. Biomarkers in cervical cancer screening. Dis Markers. 2007;23(4):315-330.]. Waiting for further disease progression/regression after positive results to verify the effectiveness of DS does not align with current clinical guidelines, including those followed by the analyzed Center. It was also not the aim of our study, which was retrospective in nature, as clearly stated in most subsections of the manuscript. A DS-positive result necessitates colposcopy with biopsy, and in some protocols, mandatory endocervical sampling. DS is not intended to directly identify transformative infections but to immunocytochemically signal their increased risk. Therefore, it was impossible to collect data on disease progression or regression, as all patients in our study were treated according to standard clinical guidelines based on their initial results. It is an additional strength of this investigation that data derived from a real-life screening rather than from a clinical trial.

Regarding the prevalence of HPV 18 and HPV HR12 in the LSIL/CIN1 group, our data indeed highlight their significance. Current guidelines recommend colposcopy for HPV 18+ cases regardless of cytology, while HPV HR12+ cases with NILM or, under certain conditions, ASC-US and LSIL can be observed for 12 months. Thus, even limited genotyping for types 16/18 has significant clinical value in determining whether patients should undergo further diagnostic procedures or surveillance, even if post-colposcopy results indicate LSIL/CIN1. Since HPV 18 is associated with a higher risk of progression to HSIL, this information is crucial for patient management in subsequent screening rounds. To further clarify the natural history of HPV, including the role of transforming infections in high-grade cervical precancers due to persistent HPV infection and the production of viral E6 and E7 oncoproteins that mediate oncogenic transformation by disrupting cell cycle regulation, we have added additional lines in the Introduction section along with corresponding references. Thank you again for your comment.

Comments 4: Perhaps a more detailed representation of information about patients and samples in the Materials and Methods section (i.e., duration of sample storage, detailed age stratification (since the expression of p16ink4a increases with age), HPV viral loads (or, at least, HPV-specific signal intensities), cytological findings, etc) would allow the authors to identify additional patterns in the distribution of the results of different analyses by subgroups of patients. I.e., importantly, no significant difference in the prevalence of HPV 16  and DS-positivity rate was found between negative and LSIL/CIN1 groups. But was there concordance between DS-positivity and HPV16 positivity? Have the authors identified DS-positive but HPV-negative cases?

Response 4: We would like to thank Reviewer 1 for these suggestions. The duration of sample storage is detailed in lines 120–122, and the overall age stratification, presented as the number of patients per group, is provided in lines 232–234 of the manuscript. However, the <25 and >50 age groups with DS results were too small to reach statistical significance compared to the other age groups, so DS age stratification was not analyzed.

The analyzed material consists of real clinical data rather than experimental samples. The HR-HPV assays used in the study are among the most well-validated (all listed on the current ESGO 2024 list). These assays are reported by laboratories as either positive or negative, without providing information on viral loads or HPV-specific signal intensities. Additionally, we did not analyze cytological findings independently. The current recommendations (Polish 2021, ASCCP 2024) allow the use of DS as a secondary reflex testing for indicated HR-HPV-positive cases. Therefore, DS results were not analyzed in relation to cytology outcomes in this study. We would like to emphasize once again that this study was a retrospective analysis of screening data, which strictly adhered to the recommendations in force at the time the study was conducted.

As Reviewer 1 noted, there is no statistically significant difference in the HPV 16 prevalence or DS-positivity between the negative and LSIL/CIN1 groups, which is expected (and for the first time this has been investigated in the population of this region), as HPV 16 is more commonly associated with HSIL/CIN2+ lesions. Most LSIL/CIN1 cases tend to regress spontaneously, so the lack of statistical difference does not diminish the clinical relevance of this finding.

While we did not independently analyze correlations between DS and HPV 16 status outside of histological outcomes, we plan to address this in a forthcoming paper. In this study, our focus was on the correlation of screening results with histological findings.

We did identify a few DS-positive but HPV-negative cases. These patients were referred for DS based on positive cytology results (in the primary cotesting group) or were a part of a small group of patients who underwent cotesting PLUS (cotesting with DS regardless of HR-HPV and cytology results), as they met the inclusion criteria for the final study group.

Comments 5: In my opinion, of particular interest is the heterogeneity of "histologically negative" patients with respect to their DS status and the high proportion of DS-positives in this cohort. However, this group of patients and this result is discussed quite superficially in the manuscript. There is a lack of detailed information about the reasons for referring patients from these subgroups for colposcopy and biopsy, about previous cytological conclusions. It is possible that DS-positive and DS-negative patients from this cohort differed among themselves and in other characteristics. Also to my view, it could be interesting for the reader to analyze the DS status of the samples for which discrepancies were obtained between different methods of testing, i.e. between cytology, histology, and colposcopic findings. Such an analysis could help point to possible causes and sources of discrepancies.

Response 5: We thank Reviewer 1 for another insightful comment. As outlined in the Materials and Methods section, our study is a retrospective analysis. DS testing was performed for all HR-HPV-positive patients, and additional cotesting plus was conducted upon patient request. In Responses 3 and 4, we explained the clinical role of DS as a HSIL risk biomarker and the typical indications for its use.

In this particular study, we do not provide a separate analysis of the correlation between cytology and DS results, as these correlations are part of a larger study that we plan to publish in the near future. Regarding referrals for colposcopy, all patients in this cohort were referred based on one of the following: a DS-positive result, an HPV 16/18-positive result, an ASC-US/LSIL HPV HR12-positive result, or ASC-H/HSIL cytological findings, as described in lines 154–159 of the manuscript. It is important to emphasize that the relatively high proportion of DS-positive patients with negative histological findings reflects the fact that a DS-positive result alone indicates colposcopy with biopsy, irrespective of other screening tests results. These patients may not have developed active lesions yet but require close monitoring in future screening rounds to detect potential progression. We have clarified the reasons for referring patients from these subgroups in the discussion (lines 287–289).

We agree that analyzing DS status in cases with discrepancies between screening methods would be of great interest. However, performing such an analysis would require evaluating all stages of cervical cancer screening — virological, cytological, immunocytochemical, colposcopic, and histological correlations — which falls beyond the scope of this manuscript. Nonetheless, we recognize the value of this approach and may explore it in future research.

Comments 6: At Figure 2, section a) looks redundant, since the same data are presented in section b).

Response 6: We thank Reviewer 1 for noting the similarity between sections a) and b) of Figure 2. However, section a) provides percentage values for individual histological results, which we believe enhances the clarity and interpretability of the findings. Combining these elements into a single figure might compromise readability. Therefore, we suggest retaining both sections for clarity, but we remain open to further revisions if deemed necessary.

Comments 7: The text contains a number of easily removable typos and stylistic errors, i.e., “The most prevalent HR-HPV among women with LSIL/CIN1 result were HR12” (while there is no HPV variety like HR12).

Response 7: We thank Reviewer 1 for highlighting this issue and apologize for any confusion. The abbreviation "HR12" is commonly used to refer to 12 non-16/non-18 high-risk HPV types, as outlined in the terminology used by the American Society for Colposcopy and Cervical Pathology (ASCCP) recommendations [Clarke MA, Wentzensen N, Perkins RB, et al. Recommendations for Use of p16/Ki67 Dual Stain for Management of Individuals Testing Positive for Human Papillomavirus. J Low Genit Tract Dis. 2024;28(2):124-130. doi:10.1097/LGT.0000000000000802]. Nonetheless, we have thoroughly reviewed the manuscript to correct any remaining typos and stylistic errors.

Comments 8: My general impression - the article can be published after some revision, including more detailed representation of results as well as reformulating some of the conclusions.

Response 8: We would like to thank Reviewer 1 for the positive overall assessment of our manuscript. We believe the revisions we have made address the concerns raised, and we are grateful for the constructive feedback throughout the review process.

3. Response to Comments on the Quality of English Language

Point 1: I have no comments on the language.

Response 1: We would like to thank Reviewer 1 for this comment.

Reviewer 2 Report

Comments and Suggestions for Authors

Dear authors,

Your study highlights the importance of incorporating novel biomarkers for high-grade cervical lesions, such as p16/Ki67 dual staining (DS), alongside HPV typing in the detection of precancerous lesions.

General comment:

The manuscript is well-structured and satisfactory in terms of content. However, I consider that several minor revisions should be made, particularly regarding the titles of the figures and tables to improve clarity and precision.

Minor comments:

1. Title: The term "DS" is not sufficient. Please use the full term "p16/Ki67 dual-stain" in the title.

2. Introduction: Ensure that all abbreviations are defined at first mention (e.g., IARC, LSIL, HSIL, ASCUS, ASC-H). Please check and verify that all abbreviations throughout the manuscript follow this rule.

3. Line 121-130:

a) Clarify the genotyping capabilities of the Onclarity test and Alinity m HR HPV Assay. Please explain that both Onclarity and Alinity offer the ability to individually genotype HPV 16 and HPV 18, in addition to providing pooled results for 12 other high-risk HPV types.

b) If multiple HR-HPV assays were used in the study, the current title of paragraph 2.3, “HR-HPV testing with two assays for extended genotyping,” is imprecise. I recommend considering to a shorter and more precise title.

4. Line 175: The sentence " The final study group included 559 cases with HR-HPV and histology results" needs rephrasing for clarity. Since you later mention that 10.9% of patients in the group with negative histology were HR-HPV-negative, consider revising to: "The final study group included 559 cases with both HPV DNA test and histology results."

5. Line 177: The sentence "93.0% (520/559) of the patients were HR-HPV-positive" is unclear. Either delete the sentence or provide a clearer explanation.

6. The title of Figure 2 is unclear. It should explicitly state that it pertains to HPV-positive cases.

7.  In Figure 3, why are HPV 18 and HR-HPV labeled as "else"? Please clarify this labeling.

8. The title of Table 1 is not precise. Consider rephrasing to clearly explain that it describes the characteristics of HR-HPV infections according to histological results.

Author Response

Response to Reviewer 2 Comments

(Round 1)

1. Summary

We would like to thank very much Reviewer 2 for taking the time to review this manuscript and the valuable help. We greatly appreciate the insights, and we have carefully considered each of the comments. Please find the detailed responses below and the corresponding revisions highlighted in yellow in the re-submitted file.

2. Point-by-point response to Comments and Suggestions

Comments 0: The manuscript is well-structured and satisfactory in terms of content. However, I consider that several minor revisions should be made, particularly regarding the titles of the figures and tables to improve clarity and precision.

Response 0: We would like to thank Reviewer 2 for these comments.

Comments 1: Title: The term "DS" is not sufficient. Please use the full term "p16/Ki67 dual-stain" in the title.

Response 1: We would like to thank Reviewer 2 for pointing this out. We have changed the title accordingly.

Comments 2: Introduction: Ensure that all abbreviations are defined at first mention (e.g., IARC, LSIL, HSIL, ASCUS, ASC-H). Please check and verify that all abbreviations throughout the manuscript follow this rule.

Response 2: We would like to thank Reviewer 2 for these comments. We have checked the abbreviations in manuscript and included the necessary changes.

Comments 3: Line 121-130: 

a) Clarify the genotyping capabilities of the Onclarity test and Alinity m HR HPV Assay. Please explain that both Onclarity and Alinity offer the ability to individually genotype HPV 16 and HPV 18, in addition to providing pooled results for 12 other high-risk HPV types. 

b) If multiple HR-HPV assays were used in the study, the current title of paragraph 2.3, “HR-HPV testing with two assays for extended genotyping,” is imprecise. I recommend considering to a shorter and more precise title.

Response 3: We would like to thank Reviewer 2 for pointing this out. a) Both Onclarity and Alinity offer slightly different genotyping than Reviewer 2 suggested. We have explained the genotyping for these assays more thoroughly as suggested. b) We have also changed the title of the paragraph accordingly.

Comments 4: Line 175: The sentence " The final study group included 559 cases with HR-HPV and histology results" needs rephrasing for clarity. Since you later mention that 10.9% of patients in the group with negative histology were HR-HPV-negative, consider revising to: "The final study group included 559 cases with both HPV DNA test and histology results."

Response 4: We would like to thank Reviewer 2 for these comments. We have revised this sentence.

Comments 5: Line 177: The sentence "93.0% (520/559) of the patients were HR-HPV-positive" is unclear. Either delete the sentence or provide a clearer explanation.

Response 5:  We would like to thank Reviewer 2 for pointing this out. We have changed the sentence accordingly.

Comments 6: The title of Figure 2 is unclear. It should explicitly state that it pertains to HPV-positive cases.

Response 6: We would like to thank Reviewer 2 for this comment. However, Figure 2 considers not only HR-HPV-positive cases, but also HR-HPV-negative. Thus, we believe that the title presents properly the contents of the Figure 2.

Comments 7: In Figure 3, why are HPV 18 and HR-HPV labeled as "else"? Please clarify this labeling.

Response 7: We would like to thank Reviewer 2 for pointing this out. The hierarchical character of the analysis presented in Figure 3 is described in lines 195-198. We have added the explanation of the phrase “else” below the Figure 3.

Comments 8: The title of Table 1 is not precise. Consider rephrasing to clearly explain that it describes the characteristics of HR-HPV infections according to histological results.

Response 8:  We would like to thank Reviewer 2 for this comment. We have changed the title as suggested.

Reviewer 3 Report

Comments and Suggestions for Authors

comments to the manuscript cancers-3209045 is an interesting and well-written manuscript in which the authors evaluated how different high-risk human papillomavirus types are distributed in precancerous (early and advanced) cervical lesions for a better understanding of uterine cervical cancer . . . . Some suggestions are described below.

Indicate the cutoff point of the KI67 that is considered positive or negative,

Figure 1 indicates the % in the column.

Include the limitations of the study.

Author Response

Response to Reviewer 3 Comments

(Round 1)

1. Summary

We would like to thank very much Reviewer 3 for taking the time to review this manuscript and the valuable help. We greatly appreciate it. Please find the detailed responses below and the corresponding revisions highlighted in yellow in the re-submitted file.

2. Point-by-point response to Comments and Suggestions

Comments 0: Comments to the manuscript cancers-3209045 is an interesting and well-written manuscript in which the authors evaluated how different high-risk human papillomavirus types are distributed in precancerous (early and advanced) cervical lesions for a better understanding of uterine cervical cancer . . . . Some suggestions are described below.

Response 0: We would like to thank Reviewer 3 for these comments.

Comments 1: Indicate the cutoff point of the KI67 that is considered positive or negative.

Response 1: We would like to thank Reviewer 3 for pointing this out. The details are presented in lines 151-152 of the manuscript.

Comments 2: Figure 1 indicates the % in the column.

Response 2:  We would like to thank Reviewer 3 for this comment. We believe it refers to Table 1. We apologize for the mistake and we have added the % in the columns. Thank you once again.

Comments 3: Include the limitations of the study.

Response 3: The limitations are described in lines 333-336, we have slightly extended this section in the review process.

Round 2

Reviewer 1 Report

Comments and Suggestions for Authors

I have nothing to add to my previous comments

Comments on the Quality of English Language

I have nothing to add to my previous comments